# Improvements in Germination and Growth of Sprouts Irrigated Using Plasma Activated Water (PAW)

**Rajesh Prakash Guragain** [1,*] , **Hom Bahadur Baniya** [2,*] , **Bikash Shrestha** [1] , **Deepesh Prakash Guragain** [3] and **Deepak Prasad Subedi** [1]

1   Department of Physics, School of Science, Kathmandu University, Dhulikhel 45200, Nepal
2   Department of Physics, Amrit Campus, Tribhuvan University, Kathmandu 44600, Nepal
3   Department of Electronics and Communication, Nepal Engineering College, Pokhara University, Changunarayan 44801, Nepal
*   Correspondence: rayessprakash@gmail.com (R.P.G.); hombaniya@gmail.com (H.B.B.)

**Abstract:** The extensive use of chemical fertilizers to increase crop yields in agricultural fields has had a negative impact on the environment. To produce more food on less land and fulfill the growing global demand for food, farmers will need innovative and environmentally friendly technology. Several studies have cited the positive effects of plasma-activated water (PAW) on seeds in their research findings. This study investigates the effects of PAW on four distinct seed species: phapar (*Fagopyrum esculentum*), barley (*Hordeum vulgare*), mustard (*Brassica nigra*), and rayo (*Brassica juncea*). Deionized (DI) water was treated for 5 or 10 min using the gliding arc discharge (GAD) system, which was operated by line frequency in the air. Water analysis indicates that the physiochemical parameters (electrical conductivity, pH, nitrate, nitrite, and ammonia concentration) of PAW were significantly different from DI water. Despite exposure to GAD for a certain period of time, the temperature of DI water did not alter significantly. All calculated germination parameters were significantly enhanced for seeds treated with PAW compared to the control. In addition, they displayed a significant increase in total seedling length and exhibited greater vigor. Seeds immersed in PAW absorbed significantly more water than seeds soaked in DI water, enabling rapid water penetration into the seed and early seedling emergence. This puts plasma agriculture ahead of conventional farming methods.

**Keywords:** plasma activated water (PAW); deionized (DI); plasma agriculture; reactive oxygen-nitrogen species (RONS); physicochemical parameters; seed germination

## 1. Introduction

One of the most important elements for maximum crop output to fulfill the rising demand for food is speedy and consistent germination [1]. To date, many agronomy-related investigations have been conducted in an effort to discover potential strategies for dramatically boosting crop growth, yield, and harvesting speed. The use of chemical fertilizers to enhance agricultural output and boost plant production is a prevalent practice all over the world because results can be observed relatively quickly [2,3]. Chemical fertilizers supply the exact nutrients that plants require to develop. On top of that, they are quite economical. If used inappropriately, it has disadvantages in addition to advantages. The soil's pH balance is altered if it is used excessively. Numerous research studies have found that they affect soil microorganism species that are beneficial to plants in addition to changing the nutrient content of the soil. They could even cause greenhouse gas emissions. Even worse, it may pollute the air and water [4–10].

Farmers will require innovative technology to produce more food on less land in order to meet the rising demand for food throughout the world. Nonetheless, it should not be carried out at the expense of a vulnerable ecosystem [11,12]. Using low-temperature plasmas in agriculture has recently become a tried-and-true technique for boosting the

germination of seeds and speeding up harvesting. Its biological and medicinal significance are likewise exceedingly diverse [13]. Since it employs environmentally benign gases as its raw materials, the generation of low-temperature plasma has little to no adverse effect on the environment. Furthermore, it is economical. In most cases, the reactive species formed by the discharge result in positive changes in the sample being introduced to plasma. All of this puts plasma treatment at the forefront of addressing the agricultural revolution [14–17]. Low-temperature plasma's sole downside is that it needs a high breakdown voltage to produce discharge [18].

Plasma treatment of seed surfaces has been classified into direct and indirect plasma treatments based on the exposure of plasma to seeds [19]. Direct treatment involves subjecting seeds directly to gas discharge for a specified time. Reactive species, charged particles, and photons produced by discharge alter the exposed seeds, promoting early germination and growth, and are assumed to be the fundamental driving factors [20,21]. However, unlike in the prior case, the seed is not exposed to the plasma during indirect treatment. Distilled or deionized (DI) water is exposed to plasma for a specific period of time. DI water that has been exposed to plasma results in a physiochemical alteration that breaks seed dormancy and speeds its growth [22,23].

Several researchers have stated in their research findings that seeds can benefit from indirect plasma treatment. When air plasma and water interact, reactive oxygen and nitrogen species (RONS) are generated in the water, which results in the production of plasma-activated water (PAW). After investigating the impact of PAW on wheat seeds, Kucerova et al. concluded that pre-treating seeds with PAW has the potential to enhance sprouting and seedling growth. Significant improvements were also observed in the amount of soluble protein and photosynthetic pigment in the roots and leaves, respectively [24]. When compared to controls, PAW treatment significantly increased the viability index of seeds. It eliminated the wax from the sample surfaces and changed their hydrophobic surfaces to hydrophilic. The wettability was shown to be improved as the water contact angle decreased [25].

Many researchers have discovered that when the treatment time is increased, the concentration of nitrites, nitrates, ammonia, and electrical conductivity in deionized water increases while the pH value is found to decrease, indicating that the water becomes more acidic. Yet plasma is unable to alter $H_2O_2$ content significantly [26,27]. The concentration of RONS present and the degree of acidity in the generated PAW rely on a number of factors, including the exposure time, volume of water, discharge kind, power density of discharge, electrode type, the voltage applied across it, initial water chemistry, etc. [28,29]. Since the nitrogen concentration in PAW is significantly increased after treatment, it is frequently referred to as "plasma fertilizer" and serves as fertilizer for seeds. It lessens the drawbacks related to the use of chemical fertilizers [30]. All of this demonstrates that PAW may be used as an alternative to water for microbiological disinfection since it has a distinct chemical constitution [31,32].

Over 65% of Nepalese citizens depend on agriculture, although only about 21% of the country's land is suitable for cultivation [33,34]. It is crucial to employ alternative methods to produce more food on less land in developing nations such as Nepal without causing environmental harm in order to combat poverty, improve food security, and support sustainable agriculture [34,35]. To determine whether indirect plasma treatment can be a viable alternative in agriculture for promoting plant growth and development, a study was conducted on four seeds: phapar (*Fagopyrum esculentum*), barley (*Hordeum vulgare*), mustard (*Brassica nigra*), and rayo (*Brassica juncea*). The gliding arc discharge (GAD) system, which was run by line frequency, was used for the treatment of deionized water.

## 2. Experimental Set-Up

The detailed schematic of the GAD system used to treat DI water is shown in Figure 1. Two knife-shaped electrodes with dimensions of 80 mm in length, 25 mm in breadth, and 3 mm in thickness were created. These aluminum electrodes were mounted in a 150 mm

long, 150 mm wide, and 150 mm thick transparent polycarbonate cube with one open end, maintaining a minimum gap of 3 mm between them. A small hole was drilled into the top of the cube so that piping from a pump or gas cylinder could be inserted. Using a 13.30 kV alternating current voltage at line frequency and atmospheric pressure, a discharge was made between two electrodes in ambient air. To reduce the high current in the discharge zone, a ballast resistor with a resistance of 1.7 megaohms was placed between the series connection of the step-up transformer and one of the electrodes. With a 10-kiloohm shunt resistance, the other electrode was grounded. To monitor the voltage and current across the discharge zone, a PINTEK (28-HF) high-voltage probe and a Kenwood (PC-54) oscilloscope probe were utilized. These two probes were linked to the Tektronix (TDS-2002) oscilloscope, which displays time-varying current and voltage waveforms.

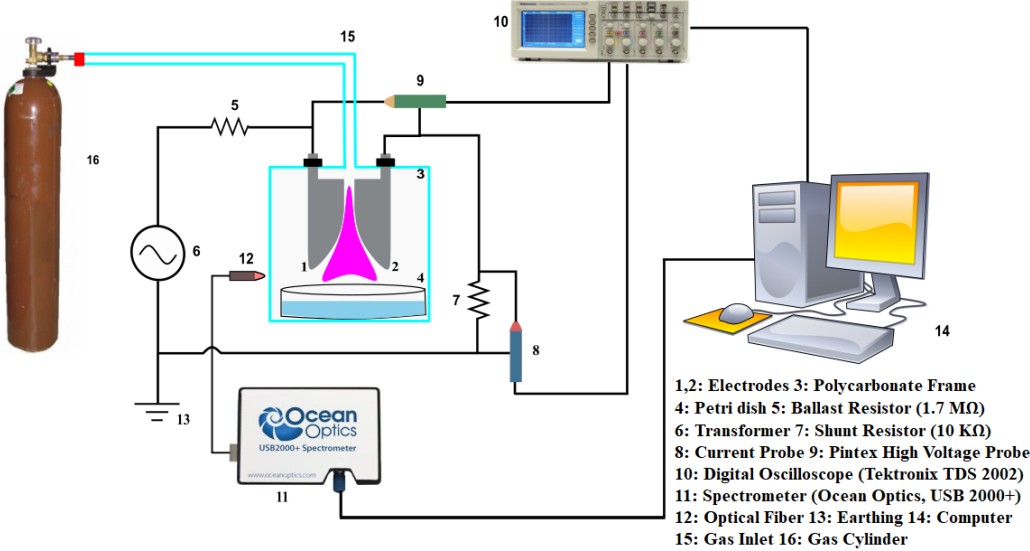

1,2: Electrodes 3: Polycarbonate Frame
4: Petri dish 5: Ballast Resistor (1.7 MΩ)
6: Transformer 7: Shunt Resistor (10 KΩ)
8: Current Probe 9: Pintex High Voltage Probe
10: Digital Oscilloscope (Tektronix TDS 2002)
11: Spectrometer (Ocean Optics, USB 2000+)
12: Optical Fiber 13: Earthing 14: Computer
15: Gas Inlet 16: Gas Cylinder

**Figure 1.** Image of the Experimental setup.

## 3. Materials and Methods

### 3.1. Water Treatment and Seed Sowing

A 100 mL beaker was filled with 40 mL of deionized water. The water was exposed to GAD produced in the air for five to ten minutes. The plasma-treated water (5 and 10 min) was immediately preserved in laboratory glass bottles and sealed with silver paper after treatment. The seeds utilized in this study were supplied by the "Nepal Agricultural Research Council (NARC), Lalitpur, Nepal." For sowing, we only chose seeds that were visibly intact. Seeds that had been visibly broken, crushed, or infected were avoided. The petri dish (100 mm in diameter and 15 mm in height) was utilized, and the base was covered with Whatman filter paper. Separately, 25 mL of deionized and plasma-treated water (5 and 10 min) were placed in a petri dish, and 50 seeds were sown in each Figure 2. Germinations were monitored at normal room temperature and pressure conditions. To compensate for evaporation loss, equal quantities of deionized and plasma-treated water were added to the relevant petri dish.

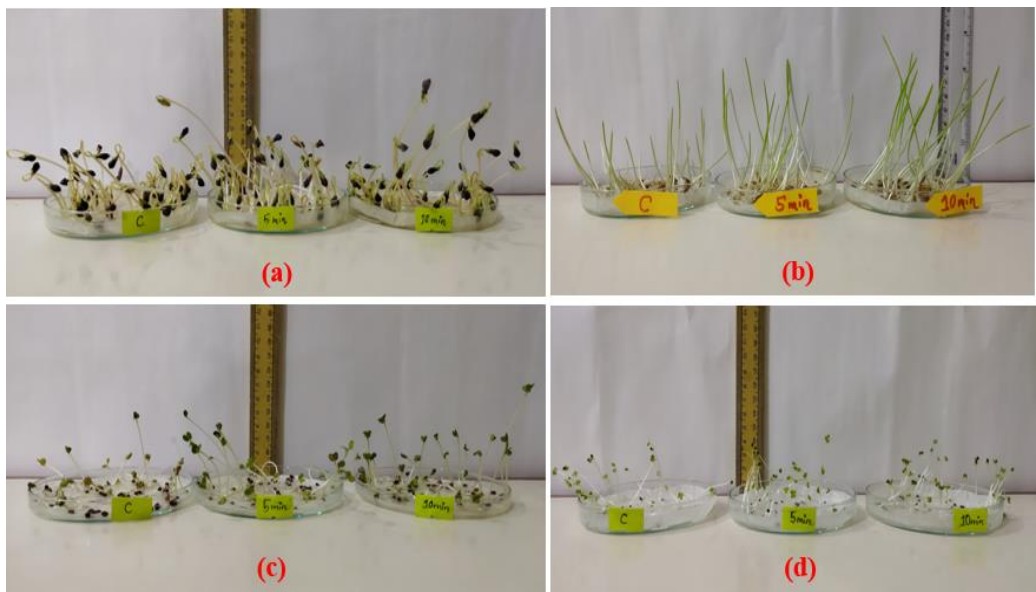

**Figure 2.** Germination of (**a**) phapar, (**b**) barley, (**c**) mustard, and (**d**) rayo seeds sown in a petri dish irrigated with DI water and PAW (5 and 10 min).

### 3.2. Germination Parameters

The number of germinated seeds was counted up to 18 days after the seed was sown. Every day until the 18th, the counting was conducted at a specific time.

### 3.2.1. Final Germination Percentage (FGP)

FGP is an estimate of the fraction of seeds that have developed into seedlings. It presents insight into the viability of a seed population [36,37].

$$\text{FGP (\%)} = \frac{\text{number of normally sprouted seeds on the 18}^{\text{th}}\text{ day}}{\text{number of seeds sown}} \times 100 \tag{1}$$

### 3.2.2. Mean Germination Time (MGT)

The MGT is an estimate of the number of days required for a seed to produce seedlings, and it relies on the day on which the majority of seeds germinate. Seeds that sprout in a short period of time after being sown will have a low MGT, and vice versa [38,39].

$$\text{MGT (day)} = \frac{\sum_{i=1}^{l} n_i t_i}{\sum_{i=1}^{l} n_i} \tag{2}$$

where, $n_i$ = number of normally sprouted seeds on the $i^{\text{th}}$ day; $n_i t_i$ = number of normally sprouted seeds during the ith period of time.

### 3.2.3. Coefficient of Variation of Germination Time ($CV_t$)

$CV_t$ enables comparisons of germination uniformity regardless of the magnitude of MGT [40].

$$CV_t \ (\% \ \text{day}^{-1}) = \frac{S_t}{\bar{t}} \times 100$$

where,

$$S_t = \sqrt{\frac{\sum\limits_{i=1}^{l} n_i (t_i - \bar{t})^2}{\sum\limits_{i=1}^{l} (n_i - 1)}} \tag{3}$$

such that $S_t$ = standard deviation of germination time and $\bar{t}$ = Mean germination time.

### 3.2.4. Coefficient of Velocity of Germination (CVG)

The coefficient of velocity of germination reveals information on the speed of germination [41,42].

$$CVG\ (\%) = \frac{\sum_{i=1}^{l} n_i}{\sum_{i=1}^{l} n_i t_i} \times 100 \tag{4}$$

### 3.2.5. Germination Index (GI)

A greater GI correlates with an increased final germination percentage (FGP) and mean germination rate (MGR) [43,44].

$$GI\ (\%\ day^{-1}) = \frac{\sum_{i=1}^{l} n_i}{t_i} \tag{5}$$

### 3.2.6. Uncertainty of Germination (U) and Synchronization of Germination (Z)

U provides information on the degree of germination dispersion. A value equal to zero signifies that germination is well coordinated [45].

Z is the measure of the extent to which sown seeds germinate at the same rate or at the same time. Its value lies between $0 \leq U \leq 1$. The value equal to one signifies that all of the sown seeds grew into seedlings at the same rate [40,45].

$$U = \sum_{i=1}^{l} f_i \log_2 f_i \ \text{ and } \ Z = \frac{\sum_{i=1}^{l} C_{n_i,2} t_i}{C_{\sum n_i,2}} \tag{6}$$

where $C_{n_i,2} = \frac{n_i\ (n_i-1)}{2}$ represents the combination of seeds sprouted in the ith time and $f_i = \frac{n_i}{\sum_{i=1}^{l} n_i}$ represents the relative frequency of germination such that $n_i$ is the number of seeds sprouted in the ith time.

### 3.2.7. Mean Daily Germination (MDG) and Germination Value (G-Value)

MDG measures the average number of seeds that germinate each day [46,47].

$$MDG = \frac{\text{final cumulative germination } (\%)}{\text{total number of intervals for final germination}} \tag{7}$$

G-value gives the overall number of seeds that sprouted when the germination rate starts to slow down [48,49]. The survival of seedlings is directly correlated with the germination value [50].

$$\text{G-value = Peak value (PV)} \times \text{mean daily germination (MDG)} \tag{8}$$

### 3.2.8. Computation of $T_{10}$, $T_{50}$, and $T_{90}$

The number of days needed for 10%, 50%, and 90% of the total sown seeds to sprout into seedlings is given by $T_{10}$, $T_{50}$, and $T_{90}$ calculations, respectively [39,40].

### 3.2.9. Water Imbibition Rate of Seeds

Water is absorbed by seeds through a process called imbibition. This process is crucial because seed germination, seed coat rupture, and seedling emergence are all triggered by imbibition pressure generated by the seed kernel [51,52]. In this work, the weight of 20 seeds was measured by weighing the Bel instrument (MG124Ai). Then, the seeds were immersed in 25 mL of DI water, PAW (5 min), and PAW (10 min) separately. Their weight was then measured every 2 h for a total of 10 h [53,54].

$$\text{Water imbibition } (\%) = \frac{\text{weight of seeds (after soaked)} - \text{weight of seeds (before soaked)}}{\text{weight of seeds (before soaked)}} \times 100 \qquad (9)$$

### 3.2.10. Estimation of Seedling Length

Water is believed to not only improve seed germination but also accelerate seedling development [55,56]. After 18 days post-sowing the seed, the seedlings were carefully removed from the petri dish, and their root and shoot lengths were measured using a standard scale.

### 3.2.11. Estimation of Vigor Index (I) and Vigor Index (II)

The primary goal of vigor testing is to make strategic choices about selecting viable seeds from seed lots. It indicates seed tolerance to a particular stress, so it is often considered a more sensitive parameter than the germination test [57,58]. One of the key elements of seed vigor is uniform seedling sprouting. Consequently, measuring seedling length or seedling dry weight represents crucial vigor measures [59,60].

$$\text{Vigor index (I)} = \frac{\text{FGP} \times \text{Seedling length (cm)}}{100} \qquad (10)$$

$$\text{Vigor index (II)} = \frac{\text{FGP} \times \text{Seedling dry weight (g)}}{100} \qquad (11)$$

### 3.3. Statistical Analysis

The experiment was repeated three times for each variety of seed. The obtained values for every parameter were expressed as a mean $\pm$ standard deviation. The three alphabets (a-c) in the graph and results of calculated parameters denote significant differences between different treatment times of particular samples analyzed using statistical tools: one-way ANOVA and Tukey's multiple comparisons at $p < 0.05$.

## 4. Results and Discussions

### 4.1. Electrical Characterization

GAD was created when a 13.30 kV ac voltage was applied between the electrodes. The generated discharge's time-varying current and voltage waveforms in an air environment are depicted in Figure 3. As can be seen from the figure, the voltage waveform resembles a saw wave, while the current waveform resembles a sinusoidal wave.

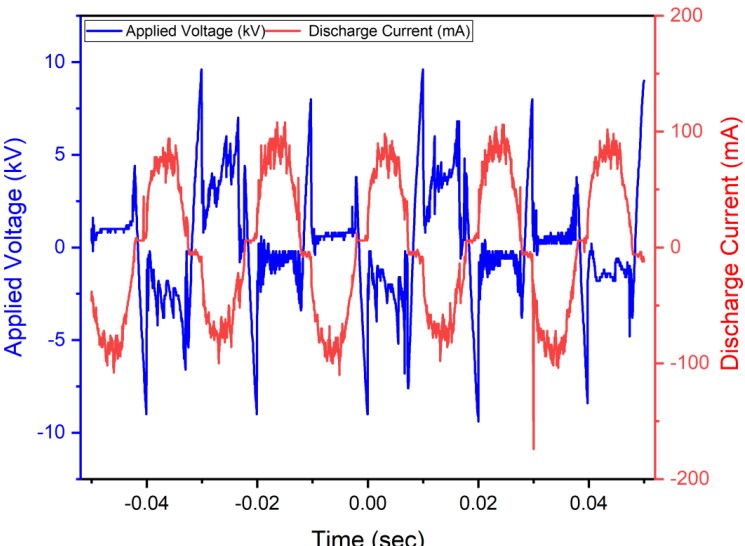

**Figure 3.** Current [I(t)] and voltage [V(t)] waveform of the generated discharge.

The power dissipated in the produced GAD is calculated by [61,62]

$$\text{Power dissipated (Watt)} = \frac{1}{T} \int_0^T I(t)V(t)dt \tag{12}$$

Here I(t) is the current, V(t) is the applied voltage to produce discharge, and T is the time period.

At an applied voltage of 13.30 kV, it was figured that the GAD system consumed 414.54 watts.

### 4.2. Optical Emission Spectra of GAD

Figure 4 depicts the optical emission spectra of GAD produced at wavelengths between 200 nm and 850 nm. Identification of the reactive species present in the generated discharge was conducted using an optical emission spectrometer (Ocean Optics, USB 2000+).

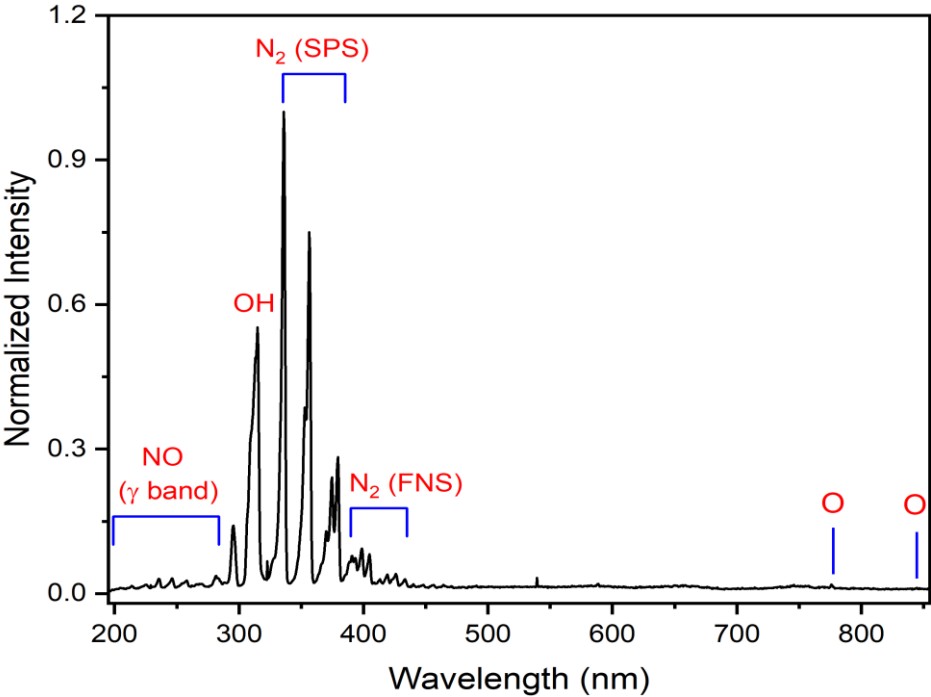

**Figure 4.** Optical emission spectrum of gliding arc discharge in the ambient air.

Low-intensity excited nitric oxide (NO$\gamma$-band) dominates the spectra at wavelengths ranging from 200 nm to 300 nm [63]. The peak at 309 nm indicates the presence of the hydroxyl radical (OH) [64]. Numerous high and low-intensity nitrogen molecules (N$_2$—the second positive system) were also detected at wavelengths spanning from 315 nm to 380 nm. Similarly, low-intensity (N$_2$ —the first negative system) is also detected at wavelengths spanning from 390 nm to 430 nm [65,66]. Singlet oxygen (O) is also present in the discharge produced, as indicated by the peaks on the emission spectra at 777 nm and 844 nm [63,67]. All of these reactive species are short-lived and produce long-lived reactive species [65,68–70].

### 4.3. Estimation of Physiochemical Parameters of DI Water and PAW

DI water was exposed to GAD for 5, 10, 15, and 20 min. The physiochemical parameters of DI water and PAW water were then measured experimentally Table 1.

**Table 1.** Physiochemical Parameters of DI water prior to and after GAD treatment.

| Physiochemical Parameters | DI Water (Control) | PAW (5 min) | PAW (10 min) | PAW (15 min) | PAW (20 min) |
|---|---|---|---|---|---|
| pH value | $6.40 \pm 0.06$ | $4.69 \pm 0.08$ | $4.48 \pm 0.05$ | $4.37 \pm 0.07$ | $4.29 \pm 0.06$ |
| Temperature (°C) | $16.5 \pm 0.02$ | $16.5 \pm 0.04$ | $16.59 \pm 0.06$ | $16.62 \pm 0.03$ | $16.64 \pm 0.04$ |
| Electrical Conductivity (µS/cm) | $0.00 \pm 0.00$ | $50.2 \pm 1.60$ | $60.2 \pm 1.20$ | $70.6 \pm 1.80$ | $76.8 \pm 1.60$ |
| Nitrite Content (mg/L) | $0.00 \pm 0.00$ | $0.49 \pm 0.04$ | $0.60 \pm 0.03$ | $0.80 \pm 0.06$ | $0.86 \pm 0.04$ |
| Nitrate Content (mg/L) | $0.00 \pm 0.00$ | $1.49 \pm 0.14$ | $4.84 \pm 0.12$ | $6.69 \pm 0.16$ | $7.20 \pm 0.12$ |
| Ammonia Content (mg/L) | $0.00 \pm 0.00$ | $1.11 \pm 0.07$ | $2.68 \pm 0.09$ | $3.01 \pm 0.07$ | $3.40 \pm 0.08$ |

Even after 20 min of plasma treatment, the temperature of DI water (control) did not change considerably. DI water is a poor conductor of electricity, but conductivity was increased after treatment with GAD. After 20 min of GAD exposure, the electrical conductivity of DI water reached $(76.8 \pm 1.60)$ µS/cm. This might be due to the addition of ions generated by the discharge. Reactive species transfer from gas-phase plasma to DI water, solvation of reactive species proceeds, and secondary reactive species are formed as a result of plasma-liquid interaction [71]. Similarly, it was observed that nitrite, nitrate, and ammonia concentrations increased linearly with treatment time. This is due to plasma-water interaction, which leads to the creation of secondary reactive species in DI water [68–70]. The pH of the DI water used in this investigation was $6.40 \pm 0.06$. However, after 5 min of exposure to GAD, it turned more acidic. The reduction in pH may be caused by the creation of peroxynitrous acids, as well as nitric acid and nitrous acid formed by the interaction of $H_2O$ molecules with $NO_x$ species [72–74].

*4.4. Germination Analysis*

FGP was calculated on the 18th day after seed germination, as shown in Figure 5. All of the Phapar seeds grew after being sown in DI water (control) and PAW (5 and 10 min), exhibiting 100% FGP. Nonetheless, barley seeds irrigated with PAW (5 min) did not differ substantially from controls, but seeds irrigated with PAW (10 min) had a 12.66% increase in FGP. Similarly, compared to the control, mustard and rayo seeds irrigated with PAW (5 min) showed no noticeable difference, but when irrigated with PAW (10 min), they showed a 16.90% and 7.59% increase in germination percentage, respectively. According to the overall findings, seeds irrigated with PAW (10-min) had an increasing seed germination percentage.

The potential benefits of plasma treatments for seed germination are mostly due to the physicochemical alteration of PAW [75,76]. If the reactive species created in PAW as a result of plasma water interaction are at low concentrations, they are considered to have a positive impact on seed physiology [73,77]. They participate in the signaling pathway and can stimulate dormancy by minimizing abscisic acid (ABA) blockage, which aids in upregulating the abscisic acid/gibberellic acid proportion, resulting in improved germination [68,78,79].

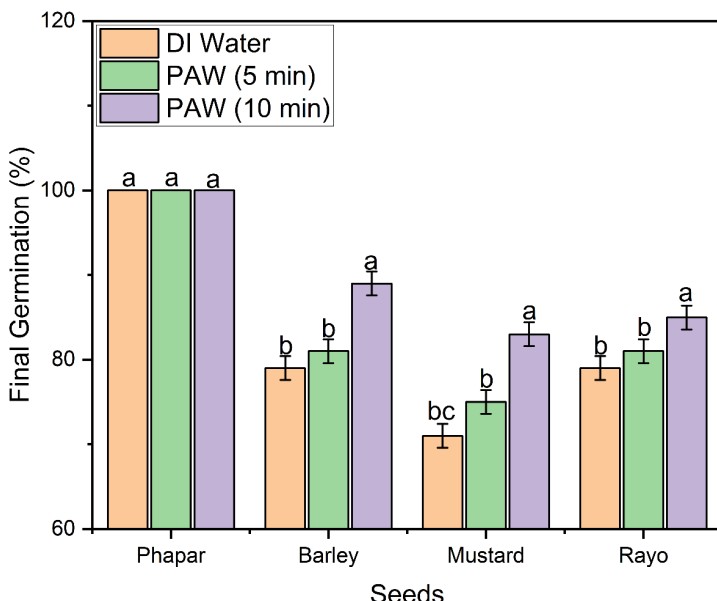

**Figure 5.** Effect of plasma-activated water on the germination of seeds. Different letters (a–c) indicate significant differences at $p < 0.05$ level among all treatments as determined by Tukey's multiple comparison test.

Comparisons were made between the MGT of control and PAW-irrigated sprouts Figure 6. The mean germination time of phapar seeds irrigated with PAW (5 and 10 min) increased by (11.35% and 16.69%), respectively, whereas mustard seeds increased by (4.42% and 3.44%), respectively, in comparison to seeds irrigated using DI water. Nonetheless, there were no discernible differences among mustard seeds irrigated with PAW. In contrast, barley seeds watered with PAW (5 min) exhibited a 7.02% increase in MGT, but those irrigated with PAW (10 min) had a 7.68% decrease in MGT. Although Rayo seeds irrigated with PAW (10 min) showed no discernible differences, PAW (5 min) increased the MGT value by 7.62%. Our MGT calculations revealed that although PAW-watered seeds sprouted more than control seeds, they germinated far more slowly.

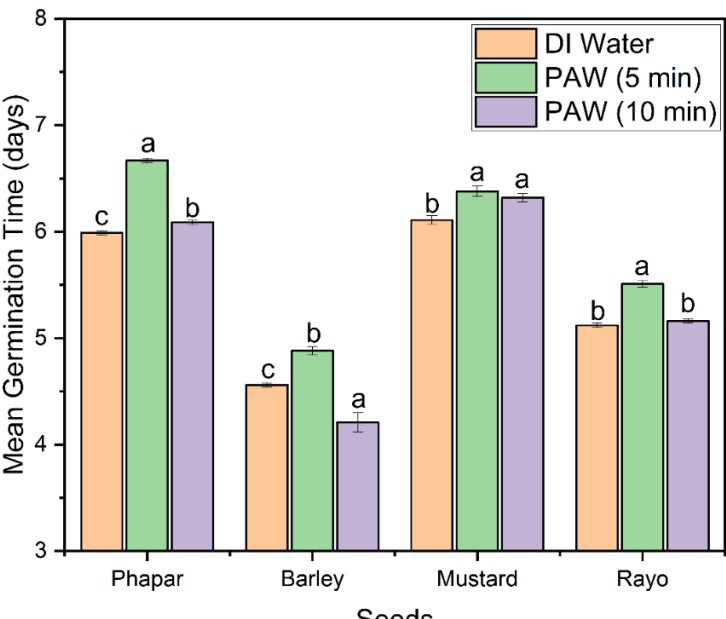

**Figure 6.** Influence of PAW on the mean germination time (MGT) of seeds. Different letters (a–c) indicate significant differences at $p < 0.05$ level among all treatments as determined by Tukey's multiple comparison test.

To compare germination uniformity, $CV_t$ values were also determined and compared in Figure 7. The phapar and barley seeds irrigated using PAW (5 min) showed an increment in $CV_t$ values of 26.61% and 108.85%, whereas seeds irrigated using PAW (10 min) showed an increment in $CV_t$ values of 25.89% and 116.01%, respectively, when compared to seeds irrigated using DI water. Similarly, mustard and rayo seeds irrigated using PAW (5 min) showed an increment in $CV_t$ values of 71.25% and 115.24%, while seeds irrigated using PAW (10 min) showed an increment in $CV_t$ values of 106.13% and 74.39%, respectively. Both phapar and barley seeds, which were irrigated using PAW (5 and 10 min), however, did not significantly differ from one another. We can conclude that irrigation using PAW improves germination uniformity.

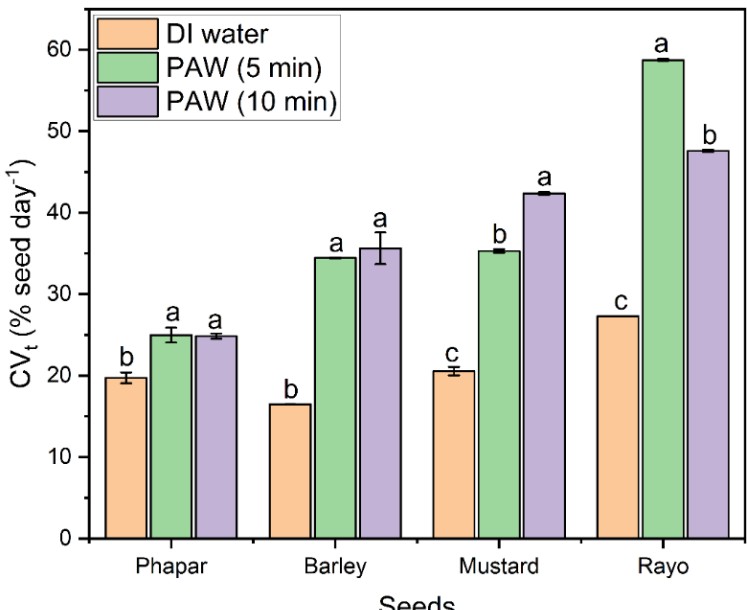

**Figure 7.** Impact of PAW on the coefficient of variation of germination time ($CV_t$) of seeds. Different letters (a–c) indicate significant differences at $p < 0.05$ level among all treatments as determined by Tukey's multiple comparison test.

The CVG of phapar, barley, and mustard seeds that were irrigated with PAW (5 min) showed a decline in value of 10.19%, 6.52%, and 4.09%, respectively, as compared to seeds irrigated using DI water Figure 8. In addition, compared to the control, the CVG of phapar, barley, and mustard seeds irrigated with PAW (10 min) showed a decline in value of 1.62%, 12.35%, and 3.18%, respectively. When rayo seeds were irrigated with PAW (5 min), the CVG value likewise decreased by 7.11%, but after being irrigated with PAW (10 min), no discernible change was noticed compared to the control. Our CVG calculations also showed, as in MGT, that even though seeds irrigated with PAW sprouted more, seeds irrigated with DI water germinated faster.

The GI values of barley, mustard, and rayo seeds irrigated using PAW (5 min) and phapar seeds irrigated using PAW (10 min) did not differ significantly from those irrigated with DI water. However, when barley, mustard, and rayo seeds were irrigated with PAW (10 min), their GI values increased by 3.94%, 20.09%, and 14.13%, respectively. In contrast, the GI value of phapar irrigated with PAW (5 min) was reduced by 7.27% Figure 9. The GI findings of our study suggested that PAW (10 min) is appropriate for seed irrigation since seeds watered with PAW (10 min) had a higher GI, suggesting a higher FGP and MGR.

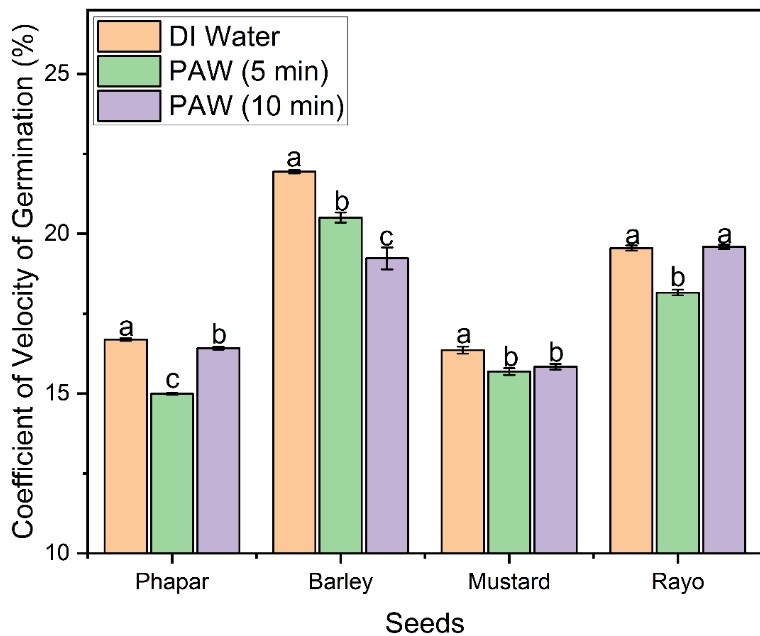

**Figure 8.** Effect of PAW on the coefficient of velocity of germination (CVG) of seeds. Different letters (a–c) indicate significant differences at $p < 0.05$ level among all treatments as determined by Tukey's multiple comparison test.

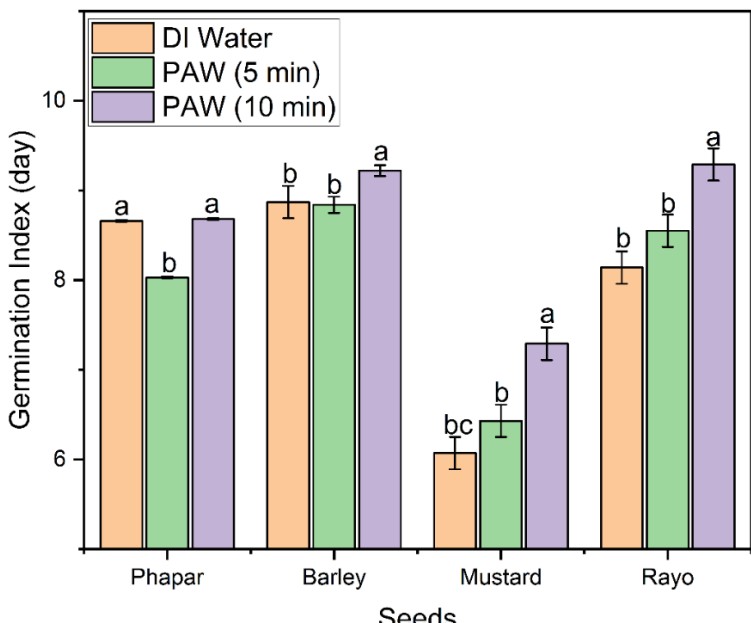

**Figure 9.** Effect of DI and PAW on the germination index (GI) of seeds. Different letters (a–c) indicate significant differences at $p < 0.05$ level among all treatments as determined by Tukey's multiple comparison test.

In comparison to seeds irrigated using DI water, the uncertainty of germination of phapar, barley, and mustard seeds irrigated with PAW (5 min) and PAW (10 min) increased by (8.96%, 18.69%, and 9.42%) and (8.96%, 48.78%, and 11.66%), respectively. There were no discernible differences between phapar and mustard seeds irrigated with PAW. In contrast, the uncertainty of germination of rayo seeds irrigated with PAW (5 min) and PAW (10 min) lowered by 14.04% and 8.19%, respectively Figure 10a.

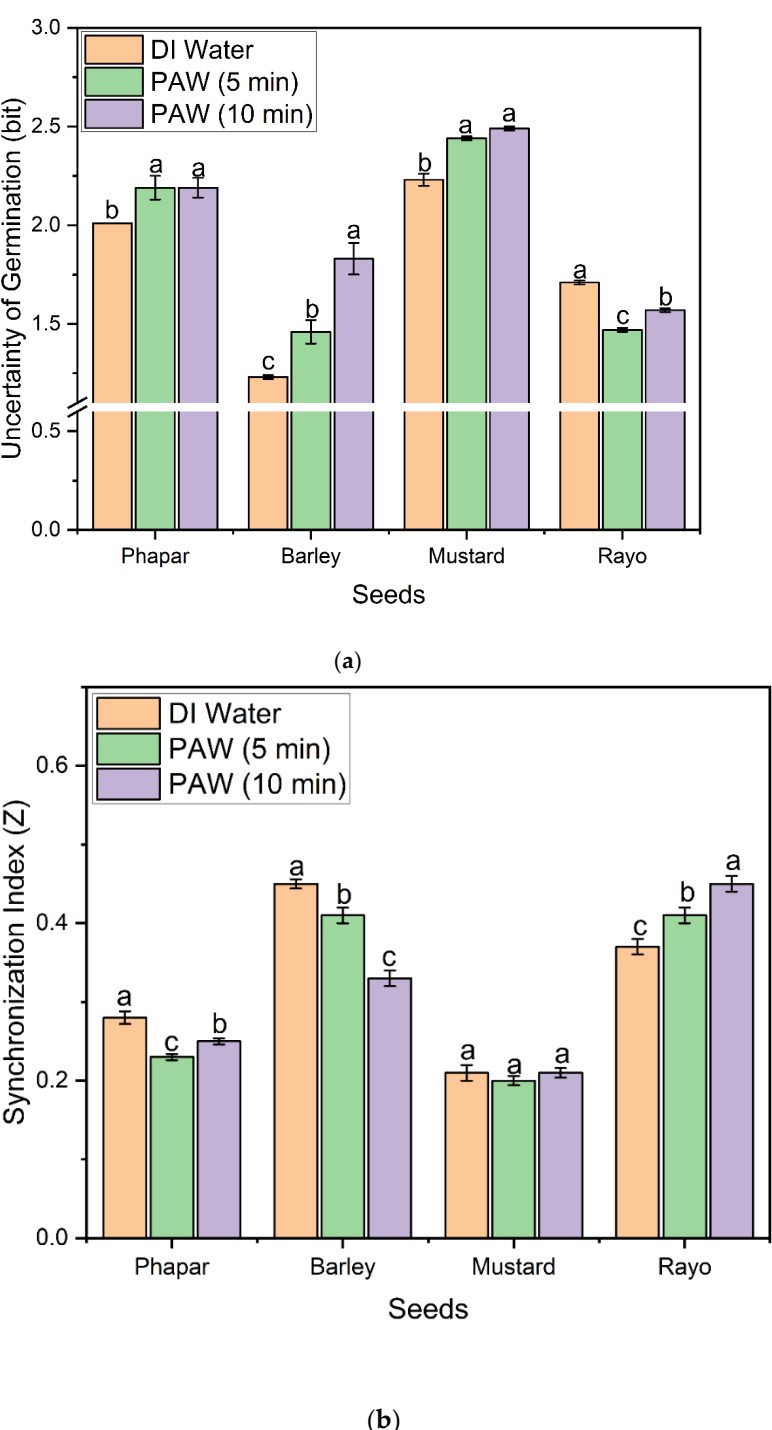

**Figure 10.** Impact of DI water and PAW on (**a**) Uncertainty in germination and (**b**) Synchronization index of seeds. Different letters (a–c) indicate significant differences at $p < 0.05$ level among all treatments as determined by Tukey's multiple comparison test.

Similarly, the synchronization of germination was also calculated and compared with the control. Compared to seeds watered with DI water, the synchronization of germination of phapar and barley seeds irrigated with PAW (5 min) and PAW (10 min) decreased by (17.86% and 8.89%) and (10.71% and 26.67%), respectively. Yet, there were no noticeable variations in the Z value of mustard seeds irrigated with PAW (5 min) and PAW (10 min) compared to the control. In contrast, the uncertainty of germination of rayo seeds irrigated with PAW (5 min) and PAW (10 min) increased by 10.81% and 21.62%, respectively Figure 10b.

These findings suggest that the seeds irrigated with PAW in our study did not germinate at the same rate, indicating that the germination process was not well-coordinated. This might be due to genetic polymorphism. Seeds can also vary greatly within a species. Furthermore, seeds from the same variety and lot may have significant variations in dormancy level or other properties. Seeds, on the other hand, vary in size, shape, internal and external structure, and water absorption ability [80–82].

The MDG values of the phapar seeds irrigated with PAW (5 and 10 min) and the barley, mustard, and rayo seeds irrigated with PAW (5 min) did not differ noticeably from the respective seeds irrigated with DI water. However, compared to the control, the MDG of barley, mustard, and rayo seeds irrigated using PAW (10 min) increased by 12.29%, 17.11%, and 7.51%, respectively Figure 11a.

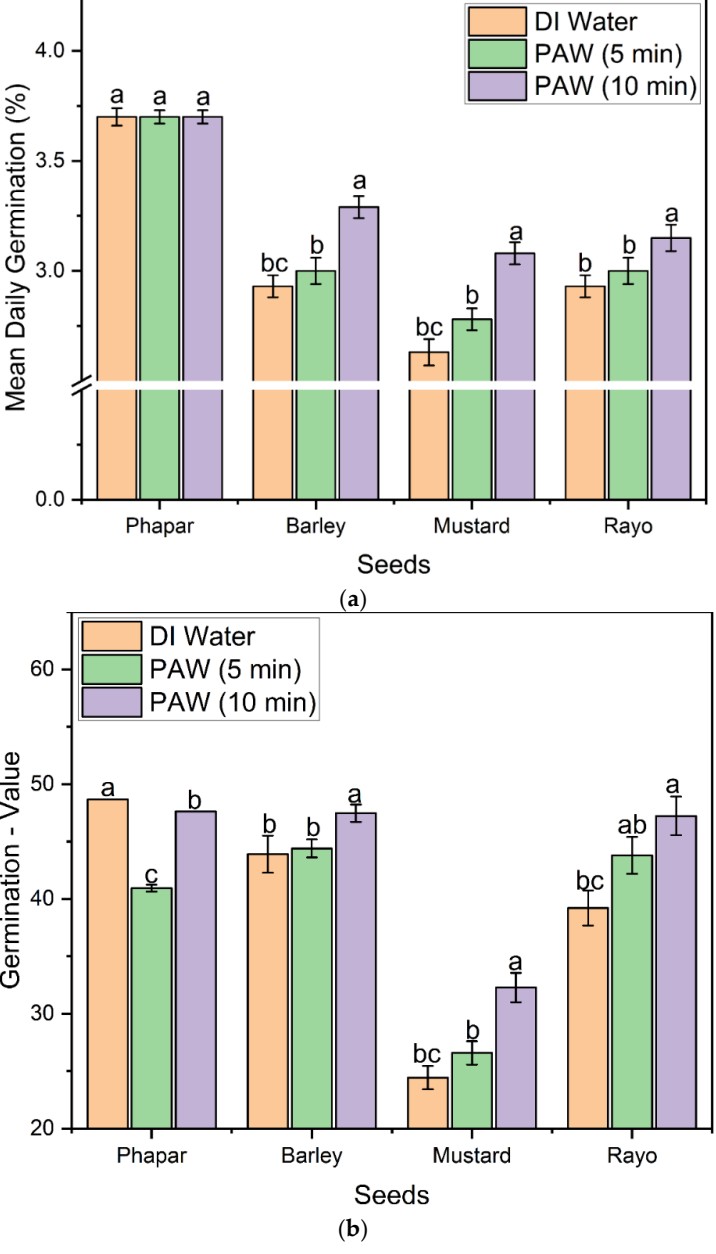

**Figure 11.** Impact of DI water and PAW on (**a**) mean daily germination (MDG) and (**b**) germination value (G-value) of seeds. Different letters (a–c) indicate significant differences at $p < 0.05$ level among all treatments as determined by Tukey's multiple comparison test.

The G-value of phapar seed was lowered by 15.88% and 2.18%, respectively, when watered with PAW (5 min) and PAW (10 min). However, when compared to seeds irrigated with DI water, barley, mustard, and rayo seeds irrigated with PAW (5 min) did not exhibit any significant improvements in G-value, while those irrigated with PAW (10 min) showed a substantial rise in G-value of (8.13%, 32.12%, and 20.42%). Our G-value data revealed that seedlings watered with PAW (10 min) boosted the total number of seeds germinated before the germination rate slowed Figure 11b. Except for Phapar, our MDG and G-value results indicated that seeds irrigated with PAW (10 min) germinated more each day, and the survival rate of the seedling was more compared to the control.

Table 2 compares the $T_{10}$, $T_{50}$, and $T_{90}$ values of seeds watered with DI water against PAW (5 and 10 min). Phapar and rayo seeds irrigated using PAW (10 min) took less time for 10%, 50%, and 90% of sowed seeds to develop into seedlings compared to seeds irrigated using DI water and PAW (5 min). Similar to seeds that were watered with DI water and PAW (10 min), barley and mustard seeds that were watered with PAW (5 min) took the least amount of time for 10%, 50%, and 90% of the seeds to sprout.

**Table 2.** Computation of $T_{10}$, $T_{50}$, and $T_{90}$ of the irrigated seeds.

| Seeds | Irrigation | $T_{10}$ (days) | $T_{50}$ (days) | $T_{90}$ (days) |
|---|---|---|---|---|
| Phapar | DI water | 4.14 ± 0.01 [a] | 5.25 ± 0.01 [b] | 6.93 ± 0.00 [c] |
| | PAW (5 min) | 4.06 ± 0.01 [b] | 6.56 ± 0.01 [a] | 8.19 ± 0.02 [a] |
| | PAW (10 min) | 3.83 ± 0.01 [c] | 5.59 ± 0.01 [b] | 7.68 ± 0.05 [b] |
| Barley | DI water | 3.19 ± 0.01 [bc] | 3.92 ± 0.01 [bc] | 4.88 ± 0.00 [bc] |
| | PAW (5 min) | 3.19 ± 0.00 [b] | 3.97 ± 0.02 [b] | 4.97 ± 0.04 [b] |
| | PAW (10 min) | 3.25 ± 0.01 [a] | 4.24 ± 0.02 [a] | 6.52 ± 0.03 [a] |
| Mustard | DI water | 3.98 ± 0.11 [bc] | 5.63 ± 0.04 [a] | 6.95 ± 0.00 [c] |
| | PAW (5 min) | 3.69 ± 0.08 [b] | 5.61 ± 0.05 [a] | 7.25 ± 0.07 [b] |
| | PAW (10 min) | 3.76 ± 0.09 [a] | 5.11 ± 0.04 [b] | 7.95 ± 0.28 [a] |
| Rayo | DI water | 3.29 ± 0.01 [a] | 4.32 ± 0.02 [a] | 6.69 ± 0.02 [b] |
| | PAW (5 min) | 3.21 ± 0.00 [b] | 4.04 ± 0.02 [b] | 8.97 ± 1.63 [b] |
| | PAW (10 min) | 3.16 ± 0.00 [c] | 3.81 ± 0.01 [c] | 7.75 ± 0.07 [b] |

Different letters (a–c) indicate significant differences at $p < 0.05$ level among all treatments as determined by Tukey's multiple comparison test.

As demonstrated in Figure 12, seeds soaked in PAW absorbed substantially more water than seeds soaked in DI water. Phapar and mustard seeds irrigated with PAW (5 min) absorbed more water than seeds irrigated with DI water and PAW (10 min).

Compared to the control, phapar seeds irrigated with PAW (5 min) absorbed 97.39% more water after 2 h of soaking, whereas mustard seeds absorbed 32.57% more water after 10 h of soaking. Conversely, barley and rayo seeds watered with PAW (10 min) absorbed more water than seeds irrigated with DI water and PAW (5 min). Compared to the control, barley seeds irrigated with PAW (10 min) absorbed 33.01% more water after soaking for 4 h, whereas mustard seeds absorbed 63.55% more water after soaking for 2 h Figure 12.

Seeds are often hydrophobic, and when irrigated with PAW, the interactions of $NO_2$ and $H_2O_2$ with wax may lead to the removal of wax from the seed coat. A high concentration of reactive species present in PAW may be involved in the surface etching of the seed coat, which increases the roughness of the seeds and leads them to become more granular, as observed in SEM images by several researchers. All of these factors improve the hydrophilicity of seeds, allowing them to absorb more water and quicker water penetration into the seed during imbibition, which accelerates seed germination by stimulating the growth of the hypocotyl and radicle [83–86].

Compared to the control, the shoot length of phapar and root and shoot length of mustard irrigated with PAW (5 and 10 min) increased significantly, although the root length of phapar irrigated with PAW did not differ significantly.

Similarly, the root and shoot length of barley seeds irrigated with PAW (5 and 10 min) and PAW (10 min) increased significantly relative to the control. However, the barley seeds treated with PAW (5 min) did not exhibit a significant increase in shoot length. There were no significant variations in root length between rayo seeds irrigated with DI water, PAW (5 min), and PAW (10 min) Figure 13. However, rayo seeds irrigated with PAW (10 min) had significantly longer shoots than the control.

The overall findings of our study showed that seeds irrigated with PAW had greater seedling lengths than the control. Nitrogen is thought to be the cornerstone of all metabolic processes that have a direct impact on plant growth. PAW is sometimes described as a "plasma fertilizer" due to its high nitrate and nitrite content, which may be used as an alternative nitrogen source [28,30,87]. In our investigation, we also found a significant rise in nitrate and nitrite levels in both PAW (5 min) and PAW (10 min) (10 min). When seeds are irrigated with PAW, it is thought to be the determining factor in better plant development. Plasma fertilizer is an ecologically benign alternative to chemical fertilizers that also offers the precise nutrients required by plants to develop [88–90].

Measuring the vigor index through seedling length (Vigor I) indicated that phapar, barley, and mustard seeds watered with PAW are more viable than seeds irrigated with DI water. Furthermore, our results reveal that phapar, barley, and mustard seeds irrigated with PAW (10 min) have higher vigor than respective seeds irrigated with PAW (5 min) and DI water. Rayo seeds irrigated with PAW (5 min) and PAW (10 min) were found to be equally viable as the control Figure 14a.

Measuring the vigor index through the dry weight of seedlings (Vigor index II) indicated that phapar and mustard seeds watered with PAW (5 min) were less viable than the control. In comparison to their respective controls, phapar irrigated with PAW (10 min) and barley seeds irrigated with PAW (5 min) were found to be equally viable, while barley and mustard seeds irrigated with PAW (10 min) were found to exhibit high vigor Figure 14b. Rayo seeds irrigated with PAW (5 min) and PAW (10 min) were found to be equally viable as the control.

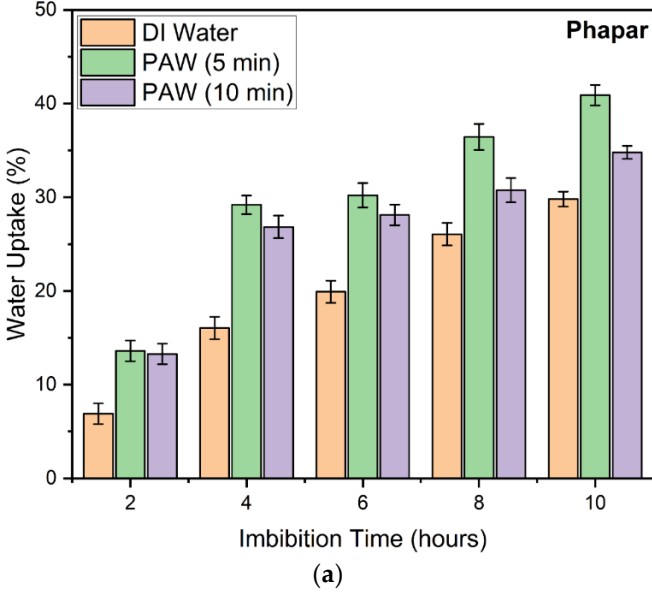

(**a**)

**Figure 12.** *Cont.*

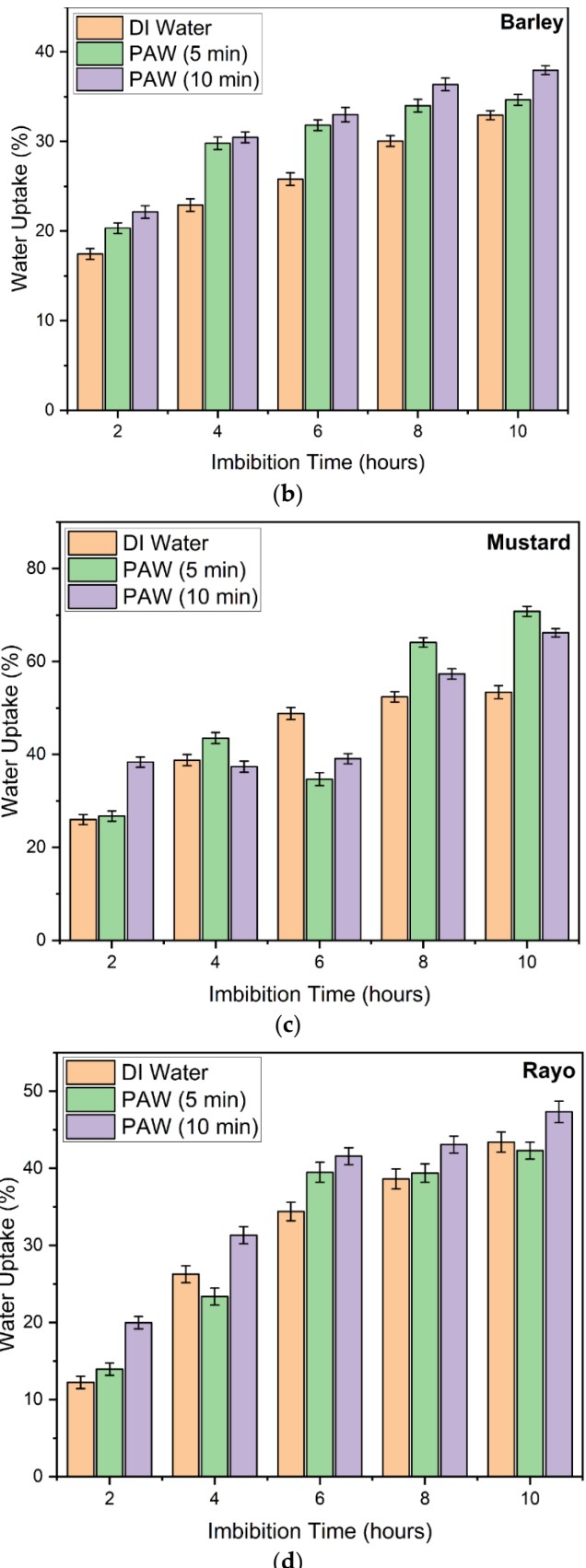

**Figure 12.** Water uptake (%) by (**a**) phapar, (**b**) barley, (**c**) mustard, and (**d**) rayo seeds. Different letters (a–c) indicate significant differences at $p < 0.05$ level among all treatments as determined by Tukey's multiple comparison test.

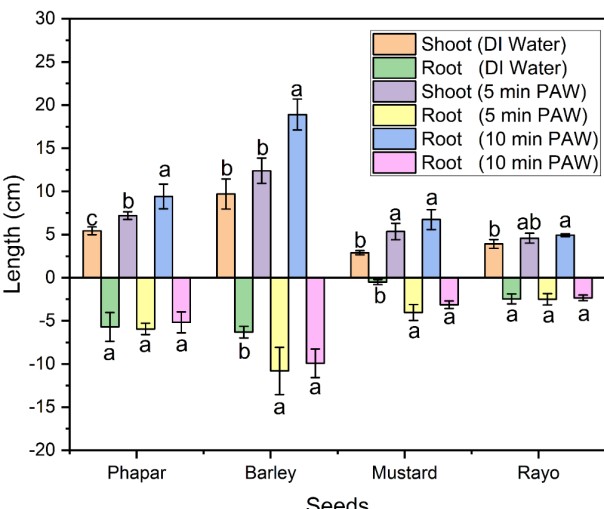

**Figure 13.** Shoot and root length of seedlings irrigated with DI water and PAW. Different letters (a–c) indicate significant differences at $p < 0.05$ level among all treatments as determined by Tukey's multiple comparison test.

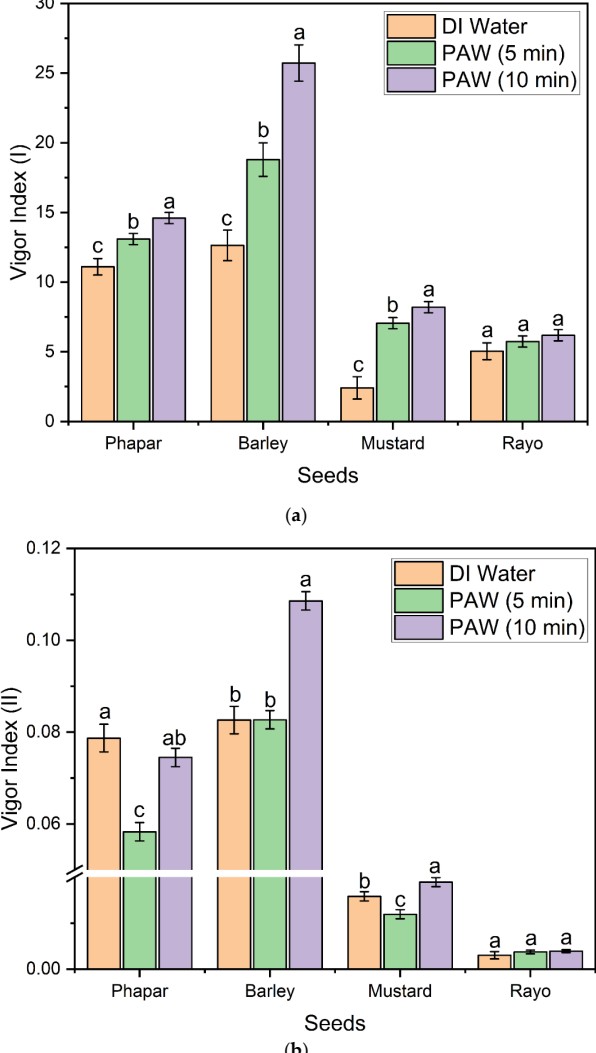

**Figure 14.** Estimation of (**a**) Vigor I and (**b**) Vigor II of seeds. Different letters (a–c) indicate significant differences at $p < 0.05$ level among all treatments as determined by Tukey's multiple comparison test.

The vigor index (I and II) is affected by (FGP and seedling length) and (FGP and dry weight) (Equations (10) and (11)). In this study, all three parameters, i.e., FGP, seedling length, and dry weight, were found to be comparatively higher in seeds irrigated with PAW than in controls. This leads to a substantial increase in (Vigor I and II), making PAW-irrigated seeds more viable. The extent to which seed physiology is altered by reactive species depends entirely on the dynamic equilibrium between their generation and metabolism. The formation of reactive species at low concentrations is considered to have positive effects on seeds, while excessive concentrations can be toxic. The complex nature of the biological research subject makes the impact of plasma treatment (PAW) on seed physiology and sprouting vary greatly. The same treatment approaches can boost germination in certain plant species while having little or no effect on germination in others. Thus, better germination value, germination index, and germination speed seed sources have higher seedling vigor and might therefore be essential selection criteria for breeding and improving this species [50].

## 5. Conclusions and Future Scope

It is crucial to employ alternative methods in agriculture to combat poverty, improve food security, and support sustainable agriculture in developing nations such as Nepal. A study was conducted on four seeds to determine whether indirect plasma treatment can be a viable alternative in agriculture for promoting plant growth and development. A discharge was created between two electrodes in ambient air using a 13.30 kV alternating current voltage at line frequency and atmospheric pressure. The GAD system consumed 414.54 watts at this particular applied voltage. DI water was exposed to produce discharge for 5, 10, 15, and 20 min. The physiochemical parameters of DI water and PAW were then measured experimentally. It was found that when the treatment time was increased, the concentration of nitrites, nitrates, ammonia, and electrical conductivity in deionized water rose while the pH value decreased, indicating it became more acidic. However, even after 20 min of plasma treatment, the temperature of DI water (control) did not change considerably. The seeds irrigated with PAW had considerably greater germination (%), uniformity in germination, as well as higher MDG and G-value compared to the control. However, the treated seeds in our study did not have a well-coordinated germination process, as all the seeds did not sprout at the same time. In addition, water imbibition (%), as well as root and shoot length, were observed to be noticeably higher in seeds watered with PAW. The vigor (I and II) analyses indicated that seeds irrigated with PAW are more vigorous. The obtained results of our study suggest that indirect plasma treatment via irrigating seeds with PAW can be considered a tried-and-true technique for boosting seed germination and speeding up harvesting.

**Author Contributions:** R.P.G.: Conceptualization, methodology, software, investigation, data curation, writing (original and final draft), writing (review and editing); H.B.B.: investigation, data curation and visualization; B.S.: investigation, data curation; D.P.G.: writing (review and editing); D.P.S.: supervision. All authors have read and agreed to the published version of the manuscript.

**Funding:** This research was partially funded by the Nepal Academy of Science and Technology (NAST), Nepal, through a Ph.D. fellowship (2076/77) and a Kathmandu University-Integrated Rural Development Program/Nepal Technology Innovation Center (KU-IRDP/NTIC) grant funded by the Korea International Cooperation Agency (KOICA).

**Institutional Review Board Statement:** Not applicable.

**Informed Consent Statement:** Not applicable.

**Data Availability Statement:** The data that support the findings of this study are available from the corresponding author upon reasonable request.

**Acknowledgments:** The corresponding author wishes to thank the "Nepal Academy of Science and Technology (NAST)", the "Kathmandu University-Integrated Rural Development Project (KU-IRDP)/Nepal Technology Innovation Center Project (KU-IRDP/NTIC)", and the "University Grants Commission (UGC), Nepal", who helped support this research.

**Conflicts of Interest:** The authors declare no conflict of interest.

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
