# Peer review of "Improvements in Germination and Growth of Sprouts Irrigated Using Plasma Activated Water (PAW)"

_water, doi:10.3390/w15040744_

Round 1
Reviewer 1 Report
[1]. There are too many sections in the results and discussion sections of the paper,Suggest streamlining and merging.
[2]. The conclusion and outlook are written directly in the results and discussion section, which should become Section 5.
[3]. There are too many conclusions in the article and it is suggested to simplify it.
[4]. The paper measured more than a dozen indicators, what are the internal links between these indicators?
[5]. Lines 256 and 257 state that PAW also inhibits seed germination. It is suggested to explain.
Author Response
We sincerely thank the reviewers and editorial board for the careful evaluation of this manuscript. We have carefully addressed most of the questions and comments raised by the reviewer. The comments from reviewers are in black, our response to reviewers is in blue, and the changes inside the revised manuscript are in red.

Reviewer 2 Report
This manuscript discusses an interesting concept in water treatment for irrigation. I commend the authors for the tremendous amount of work they completed; however, their story could be told much more succinctly. This is especially true if, as the authors state, this is a preliminary study. The authors appear to have attempted to use every possible test on the data to find results, i.e., FGP, MGT, CVt, CVG, GI, Uncertainty of germ, Synchronization of germ, MDG and G-value, Times to germ, Imbibition rate, Shoot and root length, Vigor Index I & II. They should pick the most relevant which demonstrate the effects of the Plasma water on germination and seedling growth.
Specific edits/comments/suggestions:
Line 17. … treated for 5 or 10 minutes …
Line 31-33. Delete the first three sentences. They do not contribute to the article.
L35. … To date, many agronomy …
L55. … of this puts plasma …
L65-66. … which breaks seed dormancy and speeds its growth …
L70. Delete the sentence starting “The terms “water of death…. This does not contribute to the discussion.
L91-104. Delete this paragraph. The focus of the paper is on germination and growth of seedlings. Nutritional importance of the crops tested is well known and not needed in this paper.
L110. Why are you calling this a preliminary study? Preliminary studies are usually exploratory and not done for publication. I think you have enough data to just call this a “study”.
L124. … could be inserted. Using a …
L138. A 100 ml beaker was filled with 40 ml deionized water. …
L139. 5 or 10 minutes. The use of and infers you treated for 5 minutes and then again for 10 minutes for a total of 15 minutes.
L143. … or infected were avoided. …
L146. … were placed in a Petri dish, …
L161. Write more definitively when possible, e.g., delete “seems to be”.
L172. … 850 nm. Identification of the reactive species present in the generated discharge was done using …
L174-175. Delete this last sentence.
L178. Low-intensity excited … (delete “Few”)
L187-204. Delete this section. The paper is about germination and growth. The chemical reactions can simply be referenced without including them in the paper.
L206. If DI water was exposed to GAD for 5, 10, 15, or 20 minutes, why did the authors only use 0, 5, or 10 minute treated water?
L223 – 230. Delete this section. The paper is about germination and growth. The chemical reactions can simply be referenced without including them in the paper.
L233. At what time was the counting done?
L239. Figure captions need more description. A figure with its caption should be able to be understood independent from the text.
L248. … minutes) had increasing seed …
L249. Delete the first sentence of this paragraph. It is a repeat of the last sentence of the previous paragraph.
L508. Delete reference to this being a preliminary study.
L505-534. Conclusion should be concise and reflect the findings of the study as they relate to the objectives of the study. The significant findings of the study are lost in the writing. The last paragraph of the conclusion could be deleted and the message of how PAW affects seed germination would not be lost.
Author Response

(The authors gave the same response as above.)

Reviewer 3 Report
The manuscript "Improvements in Germination and Growth of Seeds Irrigated Using Plasma Activated Water (PAW)" investigates the effects of PAW on the seeds of four different species: phapar (Fagopyrum esculentum), barley (Hordeum vulgare), mustard (Brassica nigra), and rayo (Brassica juncea). The study is interesting and may have useful practical applications.
The manuscript needs some improvements and the following comments aim to help to increase its quality.
The title could be slightly modified, in order to avoid the formulation "Growth of Seeds"
Line 28, keywords: replace "ntrogen" with "nitrogen"
Lines 81, 216 ... : I suggest to replace " rose " with a synonym all over the manuscript
Also, authors must replace "ml" with "mL" all over the manuscript.
Figure 1 - legend: Please correct 4: "Perti Dish" replacing with "Petri dish"
All the abbreviations should be explained at their first mention! The legend of figures and tables should be easy to read and understand, so the use of non-conventional abbreviations should be avoided.
Line 472: the weight is measured in grames, g and not gm - please correct!
The section "3. Materials and Methods" must contain also the determination of the germination parameters, including the explanation of the method and the calcul formulas, which must be transfered here from chapter "4. Results and Discussions".
Line 533: Replace "paw" with "PAW"
Author Response

(The authors gave the same response as above.)

Round 2
Reviewer 2 Report
The authors appear to have completed a thorough and complete revision based on the reviewers' comments.
Comment/suggestion:
Line 17. … treated for 5 or 10 minutes … (the way it is written appears that you treated samples for 5 minutes and again for 10 minutes instead of separate treatments of 5 or 10 minutes.)
Author Response
Thank you, Professor.
Based on your suggestion, it has been fixed in the revised manuscript and marked in red as
Deionized (DI) water was treated for 5 or 10 minutes using the gliding arc discharge (GAD) system, which was operated by line frequency in the air.
